# "Sometimes I think my frustration is the real issue": A qualitative study of parents' experiences of transformation after a parenting programme

Kathy McKay[1,2]*, Eilis Kennedy[2], Bridget Young[1]

**1** Public Health, Policy and Systems, Institute of Population Health, University of Liverpool, Liverpool, United Kingdom, **2** Tavistock and Portman NHS Foundation Trust, London, United Kingdom

* kmckay@tavi-port.nhs.uk

## Abstract

### Introduction

Parenting programmes help to alleviate conduct problems in children, but ensuring that all parents feel supported to attend, complete and learn from these programmes has proven difficult. Parents can feel overwhelmed and struggle to change their parenting. This article aims to inform the future refinement of parenting programmes by examining parents' narratives of how programmes motivated them to change and enabled them to put changes into practice.

### Method and results

Forty-two parents, most of whom had attended Incredible Years group sessions, were interviewed about their views and experiences of parenting programmes that focused on positive parenting practices. Analysis of interview transcripts drew on thematic approaches. Parents perceived that parenting programmes helped them to better understand their child and themselves and to let go of anxieties surrounding their child's behaviour. Better understanding included greater awareness of emotions and of behaviours their child could and could not control. Parents believed this awareness helped them to change the ways that they interacted with their child, which, in turn, helped them and their child to feel calmer. With greater understanding and calmness parents believed they became more able to see for themselves the changes that they could make in their parenting and everyday lives, and to feel more confident in putting these into practice.

### Discussion

By supporting parents to reflect on their own and their child's situation, parents perceived that programmes enabled them to improve interactions with their children without getting stuck in self-blame or feeling overwhelmed. Parents of children whose behaviour remained challenging believed that programmes led to beneficial changes in the way they felt about

**Data Availability Statement:** Although we removed place and person names from transcripts, these are verbatim records of in-depth interviews and contain details of the family living

arrangements and structure which could lead to the identification of some families. We would therefore invite researchers with interests in accessing the data to contact Dr Jane Petty, who is the overall research coordinator for the Tavistock's Research & Development team. Her email is j. petty@tavi-port.nhs.uk.

**Funding:** The PPC Study was funded by an NIHR Programme Grant for Applied Research Award LTC-RP-PG-0814-20001. The funders had no role in study design, data collection and analysis, decision to publish, or preparation of the manuscript.

**Competing interests:** The authors have declared that no competing interests exist.

their child's behaviours. Enhanced support for reflection by parents could potentially help more families to benefit from parenting programmes.

## Introduction

Parenting programmes, often facilitator-led and focused on enhancing parents' knowledge and skills around their child's development and behaviour, are effective interventions for children demonstrating difficult behaviours [1,2]. However, many families do not currently benefit from these programmes as some parents do not attend, while others do attend but their children's behaviour does not improve [3,4].

Research indicates that positive parenting approaches, and specific parenting programmes that incorporate aspects of these approaches, can help alleviate conduct problems [5]. While definitions of positive parenting vary, most refer to practices that are nurturing, empowering and non-violent, and involve offering children "recognition and guidance" and "the setting of boundaries" to support their development [6] (p.11). Positive parenting approaches tend to focus on improving parent-child interactions. Improved interactions are associated with parents feeling more competent when managing their child's behaviour [7]. In contrast, parents who believe they have little control over their child's 'difficult' behaviour report feelings of hopelessness and incompetence [8]. These negative feelings can be exacerbated when a parent is managing their own mental health difficulties [9].

Positive parenting practices are associated with better outcomes for children, such as decreased externalizing problem behaviour [10]. However, Furlong and McGilloway [11] argue that for parents, putting positive parenting practices into practice can be especially overwhelming when a child exhibits conduct problems. Doubt and colleagues [12] found that some parents go into 'survival' mode where they focus on getting through the day rather than finding more positive ways to interact with their child. Becoming stuck in surviving can trigger feelings of hopelessness and incompetency that can disrupt parent-child relationships long into the future. Completing a parenting programme can increase parents' feelings of competency [13], but encouraging attendance, learning and completion has proven to be difficult. Paradoxically, one of the psychological barriers to attending a parenting programme is parents' lack of confidence [4]. A parent already doubting themselves may understandably feel hesitant to attend a parenting programme, or struggle to learn from it, if they fear they are being judged.

To benefit from a parenting programme, parents also need the capacity to change, and to maintain those changes [14]. Parents may sense that their interactions with their children need to change, but not know how to change. Completing a parenting programme can help to turn a parent's sense of needing to change into actual changes by helping them learn how to turn intent into practice. Alvarez and colleagues found that parents who completed a parenting programme "were significantly less likely to have inappropriate expectations toward their child, [or] to respond less empathetically to their children" [15] (p. 180). Yet, such learning could also bring "a decrease in parental efficacy, suggesting that they learnt from the program that parenting is more difficult and demanding than expected and that they were still far from reaching adequate standards" [15] (p. 183).

Such complexities mean parenting programmes must take account of the needs of parents, as well as children. As Doubt et al., [12] have argued, it is important to understand how a parent translates learning from a programme into positive parent-child interactions. Recently,

Parry and colleagues [16] (p. 641) conducted an in-depth qualitative study of the mechanics of "the small changes" that three mothers described making after attending a parenting programme. Being part of the parenting programme helped the women in Parry's study reshape their perceptions of themselves as mothers and of their children's needs, which led to positive changes in their parenting. While focused on the rather different issue of how people adjust to the life changes that result from chronic illness, Aujoulet and colleagues work [17] found that being able to 'let go' of needing to control everything in one's life lessened people's feelings of hopelessness and allowed them to feel more empowered. By enabling parents to 'let go' of control, parenting programmes may similarly help parents of children with conduct disorder who feel hopeless dealing with their child's challenging behaviours. However, there is little other evidence to draw on regarding how parents who have attended parenting programme are enabled to make changes in both a perceptual and practical sense. Qualitative work examining the process of change from the perspectives of patients is increasingly helping in adult psychotherapy research by providing insights to enhance how therapists understand and respond to patients [18], but work of this sort in the context of parenting programmes is limited.

We sought to better understand the process of change as perceived by parents following a parenting programme. While the parents in our study collectively had experience of several parenting programmes, Incredible Years (IY) programme was the most common as our sampling was linked to this programme. Developed by Webster-Stratton, and delivered by trained facilitators, IY aims "to promote social competence and prevent, reduce, and treat aggression and related conduct problems in young children (ages 3 to 10 years)" [19] (p. 32). It is informed by cognitive social learning theory, and "teaches parents interactive play and reinforcement skills, nonviolent discipline techniques, including 'Time Out' and "ignore', logical and natural consequences, and problem-solving strategies" [19] (p. 35). This article examines parents' narratives of change in relation to IY and other parenting programmes, focussing on the ways that parents believed the programmes motivated them to change, and how they described making the changes occur in their everyday lives. Developing a parent-centred understanding of change could identify ways that programmes might enhance their support for parents, with the ultimate goal of enabling more families who do not currently benefit from parenting programmes to do so.

## Method

The data were collected as part of the Personalised Programmes for Children (PPC) project. The PPC project is a mixed-methods study and involves several workstreams. Its early workstreams examined the characteristics and experiences of parents who sought, or were perceived to require, support to alleviate their child's challenging behaviours. Its later workstreams are using the findings of the early workstreams to develop a personalised parenting programme, which will subsequently be evaluated in a pragmatic randomised controlled trial. The early workstreams of PPC comprised a quantitative phase and a qualitative phase (the current study), employing a sequential explanatory design [20] (p. 38). Specifically, data collected in the quantitative phase helped to inform the sampling for the qualitative phase. The overall aim of the quantitative and qualitative phases of the PPC study was to develop insights on how programmes led to good outcomes for some parents and poor outcomes for others. In a previous publication from the qualitative phase [21], we described parents' priorities for personalising parenting programmes and identified several changes that programmes could make to better align with the needs and preferences of individual parents. In the current article we report on the same qualitative dataset but focus specifically on how, from the perspective of parents, the current programmes altered their perceptions and enabled them to make positive changes in their parenting.

## Recruitment procedures and participant sample

Parents and caregivers were invited to participate in the PPC project if they had attended, been referred, or been considered for referral, to an Incredible Years (IY) parenting programme in three sites in south east England. To be eligible for the current study, parents and caregivers had to caring for a child who was aged between four and ten years at the time of recruitment and be showing behaviour which led children's services or schools to believe that a parenting programme might be beneficial. We use 'parent' hereafter to refer to all participants for brevity and to help preserve the anonymity of those who were not biological parents. Parents were not included if their child was not in mainstream school because of learning difficulties; had been diagnosed with autistic spectrum disorder; parents did not have a good understanding of spoken and written English or lacked capacity to give informed consent to participate. Research assistants attended the coffee morning sessions that were held before the start of each community-based IY programme to recruit parents to the quantitative phase of the PPC project. Staff in child and adolescent mental health services or local authorities also referred parents to the study, whether or not the parent eventually ended up attending a parenting programme. Participation in the project was voluntary and the research assistants reiterated that parents' decisions regarding participation would not influence the support they would receive from services. Quantitative data was collected about parents and children at three time points: just before starting IY, three-months afterwards, and six months afterwards.

At the third time point, with the help of the research assistants on the quantitative phase, the qualitative team invited a purposive sample of parents to participate in the qualitative phase of PPC, which comprised semi-structured interviews. Research assistants on the quantitative phase gave parents a brief explanation and an information sheet to read about the qualitative study. If parents agreed, their contact details were shared with the qualitative team. Qualitative researchers subsequently contacted interested parents, to discuss the study further. We used purposive sampling to access parents across three sites and who had differing experiences of referral or attendance at the recent IY. Specifically, we aimed to sample three contrasting groups of parents. These were: parents who had not attended IY or had dropped out of IY; parents who had attended IY but reported no change or worsening in their child's behaviour after IY; parents who had attended IY and reported improvement in their child's behaviour. Scores on the Parent Account of Child Symptoms (PACS) [22] a structured parental interview on children's behavioural issues formed the basis of assessing whether a child's behaviour had improved, worsened or was unchanged since the parenting programme. The PACS was part of the quantitative phase and completed before and after the recent IY group. Quantitative data from Group 1 parents indicated improved child behaviour after the IY programme. This was defined as a pre-post change in the PACS scores of 0.4 SD or greater at the 6 months post baseline follow up. Quantitative data from Group 2 indicated no change or worsening of child behaviour after the IY programme, defined as a pre-post change in PACS score of 0.2 SD or greater, at the 6 months post baseline follow up.

In total, 53 parents gave permission to be contacted and 43 were interviewed: the 10 parents who were not interviewed either could not be reached after several attempts to contact them (N = 9), or declined the interview invitation (N = 1). One interview was not audio-recorded due to a technical error, so 42 interviews were included in the analysis. From information gathered at the time of the interview, of the 42 participants, there were 36 mothers, two fathers, two grandmothers, and two other primary caregivers, caring for 27 male children and 15 female children. The children's average age was 6.7 years old (range 4–10 years). Twenty-four of the parents had a partner, and 18 were single, but the narratives indicated recent fluctuations in the relationship status of some participants over the course of the PPC project. While we did

not collect data on socio-economic status and ethnicity as part of the qualitative study, the three study sites chosen were based in communities of considerable socio-economic and ethnic diversity and expect that this diversity is reflected in our sample.

In line with the purposive sampling, the 42 participants were relatively evenly spread across the three sites: 15 from Site A, 13 from Site B, and 14 from Site C. The three contrasting groups based on experiences of referral, attendance and change in the children's PACS scores were: Group 1 (n = 14) parents whose children improved in their behaviour after IY according to PACS scores; Group 2 (n = 18) parents whose children worsened in their behaviour or did not change after IY according to PACS scores; and, Group 3 (n = 10) parents who dropped out or did not attend an IY programme. While purposive sampling created these three comparative groups, over the course of the analysis it became clear that Group 2 comprised two distinct sub-groups in terms of the experiences that parents reported in their interviews for the qualitative study. While none of the Group 2 parents had reported improvements in their children's behaviour as measured by the PACS, in their qualitative interviews, 15 of these parents talked of IY as being beneficial and of improvements in their child's behaviour (Group 2.1). The three remaining parents did not find IY to be beneficial and did not feel their child's behaviour had changed (Group 2.2). It must be noted though that every parent in the sample including Group 3 parents had attended sessions of at least one parenting programme, whether it was in the past or the IY groups included in this study.

### Data collection and analysis

As noted above, parents' narratives were collected using semi-structured interviews (see appendix). Most interviews were conducted by two qualitative researchers, KM and SP: KM has an extensive background in qualitative research in mental health and has received and delivered training in this area, while SP has a background in health-related research and has received training in qualitative methods. BY and KM developed the interview guides and refined these based on the ongoing interviews and analysis. Qualitative researchers began the interviews by asking the parent to describe their child and, to maintain a positive frame, usually ended by asking parents to describe 'a good day'. Questions were open-ended and structured around family life prior to IY, how and why parents had been referred to IY, or about their perceptions of parenting programmes more broadly for those parents who had not been referred. We also explored parents' experiences of attending IY (where applicable), including any perceived benefits and what they would change about the programme, other support they had accessed, and whether family life had changed since IY, or over the equivalent timescale for those in Group 3. Interviews usually took place 2–3 weeks after being contacted by the qualitative team and lasted between 16 minutes to 94 minutes, with an average duration of 47 minutes. Parents were able to choose where the interview took place and most chose their own home, with five participants choosing a café or a National Health Service clinic premises. The two researchers were present for most interviews, with one researcher leading the interview and the other clarifying any details that may have been missed. All interviews were audio-recorded, transcribed verbatim, then checked for accuracy and anonymised. Field notes were also written after each interview by the lead interviewer to record detail of the interview context and nuance which helped to inform the analysis. Group 1 reached data saturation first, after which only participants who fit the Group 2 or 3 criteria were recruited until both reached saturation.

KM led the analysis framing it through a phenomenological lens and drawing on Gallagher's "first person point of view" [23] (p. 7). We chose this to privilege the perspectives of parents and their experiences of parenting programmes [24]. This analytical perspective also

acknowledged the 'outsider' status of study team members, not all of whom were parents and none of whom had personal experience of parenting children with the type of behaviour problems exhibited by the children in the study. KM read the transcripts several times and coded them by drawing on thematic analysis [25]. The analytical process was a reflexive one where KM moved between and within participant narratives, interview field notes, and relevant literature, questioning her presumptions [26]. BY and EK, each read a sub-set of the transcripts and checked the themes over the course of the analysis. Different transcripts were also discussed during regular meetings involving the wider team of PPC investigators and researchers (RS, EK, MD, JH, SP, BY, KM) to further interrogate the data and facilitate the robustness of the analysis. The changes examined in this paper came from parents talking about their child's behaviour and their experiences of the parenting programme. This was at times in response to a direct question about change, but other times parents' descriptions of change arose in response to a different question altogether.

## Ethics

Ethical approval for the PPC project was provided by the UK Health Research Authority and the Hampstead Research Ethics Committee, London (N-434-525). Given that the study took place in the wider context of services in which some parents may have felt that they had little choice but to accept a referral to a parenting programme interviewers particularly emphasised the voluntary nature of this study. Prior to the interview formally starting, interviewers also discussed data anonymization and use of pseudonyms; all participants signed a consent form. Regardless of how long the interview lasted, each participant received £200 worth of vouchers to demonstrate respect for parents' perspectives and to recognise the time they gave to the study. Many parents we approached to be part of the PPC study were from underserved communities in terms of access to services as well as research. We anticipated that many of the parents may not feel they were not listened to or valued sufficiently, and we wanted to create a safe space for them to talk. We felt offering a valuable incentive would indicate how much we respected and wanted to hear parents' perspectives, and that we valued the time they took to speak to us, when time was a particularly stretched commodity for them. In line with the study's focus on voluntary participation, great care was taken to ensure the vouchers were not presented in a way that might have been coercive or in any way affected what parents felt they could say. In particular, the researchers did not mention the value of the vouchers while organising the interview and this was only discussed after the parent had read through the information sheet and signed the consent form. For almost all parents, the value of the vouchers was a complete surprise. Several participants spoke about the cathartic nature of the interview when it concluded.

## Results

### Context about the sample for interpreting parents' narratives

We sampled parents for contrasting reports of referral or outcomes of attendance at a recent IY programme. However, as noted above, parents were rather less distinct in terms of the perceptions and experiences of parenting programmes as described in their qualitative interviews compared to the outcomes in quantitative data that informed the sampling. It is important that we outline this context about our sample before presenting the qualitative findings that are the focus of the current paper. Specifically, although we originally sampled Group 2 parents to access parents whose quantitative reports indicated no improvement or worsening of child behaviour following the recent IY programme (indicated by PACS scores), in their qualitative interviews only three of the eighteen parents in this group talked of IY as not being beneficial

in that they did not describe any improvements in their child's behaviour. Furthermore, many parents, including those in Group 3 (which comprised parents who had dropped out or did not attend the recent IY programme), described positive experiences of previous parenting programmes, including previous IY. Indeed, many parents in in each of the groups spoke in their interviews about either the recent IY programme or a previous parenting programme as transformative, describing how their perceptions of themselves and their child had changed since the parenting programme and that they felt better about themselves and their child. it was mostly parents from Groups 1 and 2.1 who described the recent IY programme that they had attended during the PPC project as the source of the positive changes they perceived. In contrast, parents in Groups 2.2 and 3 mostly referred to other parenting programmes as the source of any positive changes they perceived. Additionally, only parents in Groups 1 and 2.1 spoke about better understanding their child as one of the positive changes they had experienced.

In the sections below, we refer to the quantitative data to provide context on the sample for interpreting the qualitative findings. In this article we primarily define change in terms of the qualitative data, that is, parents' narratives about how their perceptions of changed after participating in the programmes, and the practical changes they described making in their parenting. We present all quotes with the parent's pseudonym and group number in brackets, and organise parents' narratives within the following themes and sub-themes:

1. Better understanding themselves

    a. Reflecting on actions and decision-making

    b. Who has to change?

    c. Learning from the past

2. Better understanding their child

    a. What could be controlled?

    b. Changes made to support the child

3. Calmer parent, calmer child

## Better understanding themselves

This theme encompassed parents' reflections on perceived improvements in their awareness of their own emotions and behaviours, both past and present, and how these had affected the relationship with their child.

**a. Reflecting on actions and decision-making.** When speaking about a parenting programme, parents often described learning about themselves through reflecting on how they acted or why they made certain decisions. Some parents in Groups 1 and 2.1 described having realised that what was 'normal' in parenting had changed between raising children of different generations, or in different countries: "And it's different styles as well, you know, whereas. . .. when I brought my kids up, it was a lot more lax and lenient" (Gloria—2.1). In other cases, it did not take a generation or change of country for a parent to see how they could create a new 'normal' for their children. For example, a parent spoke about the 'tool bag' she and her husband perceived they now had to draw on for their younger son, as a result of attending previous parenting programmes for their oldest son: "And now that I've got all these other rules and thoughts in my head and tools–as I say, that tool bag–like, yes, my seven-year-old's going to have a slightly better future" (Lucy—2.2). While Lucy felt more confident parenting her

older son as well, she added that her younger son would receive the most benefit as he was going to have a longer exposure to "parents who know parenting now".

Some parents reflected that since attending the programmes they realised how they reacted to their children's behaviours was more based in frustration with themselves than with their child's behaviour. These parents described their efforts to be more empathic in their interactions with their children, trying to walk 'in their shoes', while also indicating that consistently being empathic could be difficult. Bea, a Group 1 parent, described how her daughter struggled with feelings of worthlessness and insecurity, which could worsen when her daughter felt she had disappointed her mother. She talked about an incident that had taken place the morning of the interview when her daughter had broken a mug belonging to the mother. She spoke positively about dealing with this incident and using strategies from the parenting programme–in contrast to how she might have reacted before IY, Bea remarked that she 'didn't have a fit' after the incident. More broadly, she described the steps she was taking to trying to focus on building up her daughter's self-esteem with praise and stickers. However, Bea's concern about her daughter's behaviour could, at times during her interview, present as inward-facing towards her own 'hurt feelings', rather than outwardly towards her daughter's emotional pain:

> But normally I would have had, had a fit. . .. but even though I didn't have a fit, [daughter] started saying, 'Oh I'm, I'm rubbish, I'm useless,' and I'm thinking, 'oh no'. And it just started from there. . .. I need to talk to her when she comes back, I need to try and sit her down and talk to her about how it's hurting my feelings that she's doing that. Because I can't seem to, at the time, stop her from. . .. So I'm going back to the filling her up with her stickers and. . .. maybe before the course I would have had a flip, as in 'Oh for goodness sake, not my mug. With all the mugs in all the cupboards, you had to break mine.' But I didn't show, and I know I didn't, and I was calm and. . .. For me it was big. [laughs] (Bea -1)

By helping them to ground their parenting in empathy, Group 1 and 2.1 parents felt that learnings from IY and other parenting programmes had enabled them to change the way they interacted with their children. Some spoke of realising that they criticised their children more than praising them. Describing what she had learned from a home-based parenting programme, another parent reflected: "Maybe I'm too. . .judgemental?. . . . So to change all these-, not to be judgemental and to praise her more. Which I do praise her but maybe not in the right way or maybe not as much as I have to" (Anna—3). Others spoke about being "more able to kind of put myself in [daughter]'s shoes and to understand her and what she was going through. . .. Um, like if she was having a tantrum or a meltdown, to kind of understand her feelings and help her to process her feelings" (Eva—2.1). Parents described a tool introduced by IY facilitators that involved parents helping their children name their feelings. This was perceived as a 'lightbulb moment' for some parents who, until their recent experience of IY, had simply assumed that their children would be able to talk about their emotions:

> Talking about naming her feelings and asking her about her feelings. I think that helped me. . . it helped her, sorry, because I think [it] was slightly better parenting. But it also helped me because it helped me to kind of see the world through her eyes a little bit. . .. In a way that I hadn't perhaps before. (Eva—2.1)

**b. Who has to change?.** Different to empathy, but in line with increasing self-reflection, some parents remarked that they had begun to think that, in some ways, they needed to change more than their children did. One parent pointed to her growing realisation that her own reactions might be more of an issue than her daughter's behaviour: "I'm doing everything. . .. That

I can. . .. So everything available! [laughs]. . .. sometimes I think my frustration is the real issue" (Margot—2.1).

Changing was nevertheless still a work in progress in the narratives of many parents. Anna, a Group 3 parent described how she was starting to see that she had been replicating her own childhood in interactions with her child, and the way her parents had acted. While she had dropped out of the recent IY, she believed that one-on-one parenting support in her home was now helping to 'rewire her mind'. Her narratives throughout the interview suggested she was in the process of shifting her language towards her daughter from criticism to praise and from reaction to reflection:

> I call her lazy because she is! [laughs] Um, she will never clean her room, she will never tidy the room, she will never-, so I need to find a way to make her do these things without open-ing my mouth and saying the wrong things! Which is hard for me! [laughs]. . .. I need to-, as I said to her 'rewire my mind,' how to. . .Because it's not easy, being brought up as a [nationality]. It's different, we don't have the same way. I mean we have parents that they do judge us all the time! And they will say things like 'oh my god, are you stupid! What have you done here?' (Anna—3)

Another parent also realised how she communicated with her son directly affected his response, although she felt she had learned this from a prior parenting programme completed before the recent IY:

> If I'm having a bad day he'll rub off on that and it depends on how I talk to him–it's all about communication, communication's key. If I'm going to shout at him, don't expect him to talk to me back, expect him to shout at me back, you know. Um, it's learning to con-trol that, this was this other course and you know, it's taken time. Right, you're having a bad day but don't let him know that you're having a bad day. Still speak to him, still give him the choices, even if he makes a wrong ones don't go off on him, make it like child led. (Penelope—2.2)

Alongside their narratives of working through these feelings of frustration, some parents also described learning to distinguish between behaviour from their children that was danger-ous and needed to be responded to or changed, and behaviour that was merely undesirable and could be tolerated:

> And that was another thing we dealt with in the course as well, it's what needs changing and what doesn't. Is it to do with safety? Is it to do with—you know, if it's not putting any-one at risk.. . . Why put so much focus on it?. . . . But, of course, if it's something that is going to affect their safety, you know, like road safety then of course you have to do some-thing about it. (Iris—2.1)

Other parents acknowledged that while their child's undesirable behaviours could still make them feel anxious, they felt more able to let go of their urges to control their child and of the accompanying anxiety:

> I was thinking 'do I really need to understand every single thing?' No, I'd like to, it'd be nice but it's literally just let it go.. . . Just let it go. [laughs] Literally, any time I start thinking [mimes stress] I just go like this [whooshing sound], throw it away, throw it away, throw it away, it's OK. Take a deep breath. . .. Yeah, yeah, just throw it away. . . (Bryony—1)

**c. Learning from the past.** In line with a perceived increase in empathy and realisation of the importance of distinguishing between dangerous versus undesirable behaviours in their children, some parents also spoke about how past and recent experiences of IY had taught them to be kinder towards themselves which, in turn, allowed them to be kinder to their children. This was a different narrative to increasing one's empathy; here, parents talked about how programmes had helped about enhance their confidence and energy, particularly after a trauma. For example, Clara spoke of how a facilitator at a previous IY programme encouraged her to work with daily affirmations as a way to overcome insecurities. These affirmations tended to be reminders of small accomplishments, made all the more important as Clara and her family were coming up to the anniversary of a traumatic bereavement. During difficult times, Clara believed these affirmations helped her to be forgiving of herself and her children, as they struggled with their grief:

> So if I want to go and scream, I go to into the bathroom and read [the affirmations] and like 'oh yeah!'. . .. I managed to get [daughter] dressed today you know. It's silly little things. . .. And it's an achievement and. . . you have to focus on the achievements because then you. . . it's retraining your brain isn't it? [sighs/laughs] (Clara—3)

At the time she was interviewed, Olivia (a Group 1 parent) was recovering from a period of depression after a difficult set of life events, including a relationship break-up, acknowledged that her son had also been affected by her depression. Olivia believed that how she perceived herself affected her son but also recognised the importance of not blaming herself:

> Because it is just about changing your perspective of things. . .. they were saying you know about telling kids, you know if the kids hears so many negative things you need to kind of boost that and counteract that and. . . . It does make a huge difference, once you put it into practice, you can see it and you think oh my goodness! Yeah. . .. Yeah, it's being a bit more aware of how you're parenting. . . . It doesn't mean like necessarily mean that you're bad, it doesn't mean you know it's just normal we all slip up. . . (Olivia—1)

In his interview, Peter similarly reflected on learning from his recent IY experience about the need to also be forgiving of himself on the days where his ideals for the day did not turn into realities. He felt more able to see where he might have made a mistake if his son behaved in a challenging way and tried to use that as a learning for the following day:

> I am aware that I, you know, start the day out every day thinking, right, today, I'm going to not cock up, I'm not gonna make a mistake. I'm, you know, when he kicks off I'm gonna try and stay calm. . . I'm recognising when I am like doing it wrong. . . . and, um, trying to go and make sure that next, next time I get the opportunity to deal with it the way that I should deal with it that's best for him really. (Peter—2.1)

Parents also felt that kindness could change every-day interactions with children too. Since attending the programmes some parents described realising that they could often sound grumpy towards her children, even if they were not feeling that way: "Like maybe sometimes I could come across, not aggressive, but quite grumpy if I'm like [shouty grumpy tone] '[son] come on and get ready!' rather than [calm sweet tone] 'Come on [son] get ready now!' You know, it made you think about things like that" (Elodie—2.1).

Some parents believed the IY programme had helped them to reflect on how past negative experiences had shaped how they reacted to challenging behaviours from their children. A

Group 1 parent, Jac, described having learned to recognise that the anger she was carrying was hurting her relationship with her child, although this had been hard for her to accept at first:

> You have to embrace even though he's, your child's doing things to hurt you, you should not feel it. . .. So whatever [the facilitators] were teaching, it was hard for me to take it but when I took it, it was good. Because I then realised if I don't take it, then me and my [son's] relationship is finished. Because negative doesn't work properly, the positive does do good things, magical, magical things. You understand? So I took it. I took. . .I think I let the anger go. (Jac—1)

Being able to talk more calmly to her child was healing their relationship: "It taught me not to be overwhelmed over minor, petty things. Rather there are things to focus on than to be angry. When you're angry, you lose everything around you. Even the air you breathe it has become so bitter you don't understand. So I'm in love with that [laughs]" (Jac—1).

## Better understanding their child

This theme encompassed parents' reflections on perceived improvements in their awareness of their child's emotions and behaviours, what their child could control, and how changes in a parent's reactions could help to better support their child.

**a. What could be controlled?.**   Some parents from Groups 1 and 2.1 spoke about the parenting programme helping them realise that challenging behaviours were often outside their child's control. For example, instead of trying to stop her son's fidgeting, which had caused significant worry and blame before her recent IY experience, Julia described feeling more able to respond in ways that calmed her son at times, and at other times of simply accepting his need to fidget:

> But—just acknowledging that there's a lot he can't help has caused us to back off, and I found rather than constantly telling him to stop fidgeting, if he's sat next to me I just like to put a hand on his leg or something, and that seems to help him calm. . .. And then, because we haven't said anything, there's no sort of stress about it and he's. . .. Or sometimes I just let him fidget. [laughs]. . .. He seems to need to. [laughs] (Julia—2.1)

Several children in the study had learning difficulties that their parents sometimes struggled to understand. Margot spoke of finding it hard to understand why her daughter struggled with certain numbers, and of how her daughter reacted against this with frustration. For Margot, their interactions around schoolwork were becoming increasingly stressful and she believed that this heightened stress was derailing the tools and strategies Margot was trying to put in place after the parenting programme to assist her daughter's learning. While still not understanding the reason the confusion, Margot spoke of trying to avoid adding to her daughter's frustration:

> I'm doing my part. . .. Doing our times tables. . .. But she's confusing twelve and twenty all the time. . .. and sometimes I can't understand why. . .. Um, but didn't say [anything]. . . yeah it's hard to learn when she's confusing the two numbers. . .. So I feel frustrated—what can I do?. . .. Because she knows it's, um, it's twenty, she needs to put twenty, but she puts twelve. . .. But that's why she's failing the times table. (Margot—2.1)

As we outline above, many parents who participated in the study had done other parenting programmes, and their narratives indicated that some of their learning intermingled across

various programmes. For example, weaving together what she had learned at IY and a previous parenting programme, Charlotte talked of now better understanding that her son's "melt-downs" could happen unexpectedly because of his sensory issues:

> And it just hit home to me again about when he is having a meltdown, not to get stressed and angry with him, not to get angry with him. He isn't choosing necessarily to do that, but to help him manage his meltdown, to get through it. And it just sort of encouraged me again. (Charlotte—2.1)

**b. Changes made to support the child.** Parenting programmes appeared to help parents keep in mind that their children could not necessarily understand things or act in the same way as adults. Some parents spoke about learning that children often needed extra help with certain tasks, and of developing visual aids to help their child follow a routine to get ready for school without having to constantly repeat instructions. Other participants linked their recent experience of IY to their growing understanding about how and why children followed, or did not follow, instructions. Vivienne reflected that before the parenting programme, she had often become upset at her daughter's inability to follow instructions as she thought her daughter was being disobedient. After the programme, she realised it was about ensuring her daughter understood what was asked and could do whatever was being asked at her own pace and indicated that this realisation had made their interactions calmer:

> We've realised that when you ask [daughter] to do something, you can't just say, '[daughter], go and do this now' because she goes, [bored] 'yeah'. So you have to then start off earlier than you want her to do it. . . . keep on at her every. . . . five minutes and eventually it will get done but I can't expect her to just do it straight away. . .. 'Cause I was expecting immediate results. . .. So that has really helped. (Vivienne—2.1)

Ines, a Group 1 parent explained that her had son struggled to get her and her partner's attention when their younger daughter had required repeated hospitalisation after she was born. She added that it was not until the IY programme that she realised how much she and her partner had needed their son to be 'grown-up' during this time, even though he was still very young:

> The thing that was actually the best was that it reminded me he's only five. It was like, he's always been so-, because he's the first I guess he's always felt so grown-up. And we've always just thought he was a bit like-, a bit more grown-up. And then-, and we had unreal-, or I had unrealistic expectations of him and also I needed him to toe the line because we were in hospital so much. But then he needed to also be little. . ... So it's actually changed the whole family, not just me. (Ines—1)

Ines elaborated that she was now beginning to have days when she was able to respond flexibly, letting her son make decisions about what they did and also letting him be child-like. She gave an example of trying to take her son to a famous art gallery. When he ended up being more interested in the street performers outside the gallery, Ines changed her plans and 'let him lead' so they spent the day outside watching performances, rather than inside looking at art:

> And all he wanted to do was see some dude dressed up as Yoda. Like so you know those dudes who look terrible! But he's five! And eventually I was like 'you know, what, let him

do what he wants to do'. . .. I let him lead. Which I would never have done in the past. It was fabulous!. . . . But it was lovely, it was just lovely, I didn't get cross with him once. I realised when he was tired. . .. I had let go of my 'noooo, I must control your life' and he-, bless him, he was like 'adventure day was wonderful!' (Ines—1)

Bryony similarly described linking her acceptance of the need for flexibility to a recent IY programme, but in her case, it was the realisation that her eldest daughter could take on tasks around the home, responsibilities that indicated she was 'bigger' than her sisters:

She has responsibilities anyway in the house but now, um, she can put the bins out, the recycling bins are just like further, so when you come out the house she has to walk a little bit up there and she can do that. So normally I would be like 'no, you can't go outside this house without me' and she goes and she feels big. (Bryony—1)

Bryony elaborated that she struggled with anxiety, which was often related to keeping her children safe. She believed that IY had helped her to understand that her efforts to keep her children safe could be perceived as mistrust by her daughter. On a related note, Bryony also mentioned that she had started letting her daughter go to a supervised local youth group. She elaborated that her daughter now felt Bryony trusted her, which was strengthening their relationship and enabling them to share their feelings: ". . .because I am letting her do these things she's more relaxed and she's more open. . . . I suffer from anxiety so when I start panicking I let them know like 'OK, mummy's panicking. . .' Um, so it's nicer" (Bryony—1).

Some children struggled with speech and learning disabilities, as well as other physical and emotional issues. Violet had been trying out different active strategies to help their child healthily vent his emotions. Violet linked this approach to learnings from the parenting programme but also by Violet's previous experience working in a "special needs school":

'If you're feeling angry or upset, you run all the way upstairs without touching anyone, you run upstairs and you slam your door as hard as you can because that slamming of the door is the frustration'. . .. And that is what he does. . .. Even to this day. He still does that sometimes. Not a lot. But before he was doing it all the time. (Violet—1)

After slamming the door, Violet would then let her son stay in his bedroom, and choose when he was ready to come out. Indeed, this approach appeared to be so successful that they now needed to use it far less regularly:

I'd go up there and I'd be like 'Are you ready to talk now?'. . .. 'No!' 'OK' I'll leave him to it and come back and if he's says 'Yes', I'll go and sit down with him obviously and close the door behind me and sit down and I'm like 'Why did you get so upset?'. . .. And he'll kind of try and explain to me. . . . And I'll be like 'I'm going to go out now, if you're ready you can come with me. Do you want to come with me?' And he'll either come with me or he'll say 'No' . . .. It's literally leaving everything down to him. (Violet—1)

## Calmer parent, calmer child

This theme encompassed parents' reflections on their belief that they were feeling calmer since attending IY, and that this calmness was being mirrored by their children. Parents attributed this calmness to feeling more confident about their parenting since attending IY. Julia spoke about the confidence that came with knowing she and her husband were doing 'the right

things'. She added that she no longer felt stressed when a tool or strategy from the parenting programme did not work immediately and that both she and her husband could approach any challenge more calmly:

> I would say confidence was the biggest thing for me. . ... certain things we'd be doing with [son] where I was just ready to give up and think well he is never going to behave [laughs]. . . so I'll keep going and we will see if it rewards eventually. And other things that I hadn't thought to try, or I had thought about but had kind of not really felt confident enough to start with him, or things we had given up on already, actually thought we'll do what they're saying because everyone else seems to be. . .getting good results and we'll see how this goes. . ... I mean I have found just with that, I have been able to approach things a lot more calmly, which I think he is picking up on. So, the same with my husband, and then, because we're calmer, he's calmer, and everything just seems to ease down, a lot less stressful. (Julia—1)

Another parent spoke about being able to react more calmly to her daughter's meltdowns, which meant these were becoming less frequent and, when they did happen, the meltdowns were shorter. Indeed, she felt that her daughter was becoming more reflective about the times when she unable to control her anger, even if she was not always able to put some of the calming behaviours into practice:

> I've noticed that the more I remain calm, the sort of calmer she gets and, you know, I just, like I say, she's asked me like to remind her when she gets angry, about calming down and breathing and stuff. So now, yeah, when she does get sort of angry or, or in a panic, I'll just sort of say 'oh, you know, remember you asked me to remind you about your breathing and about, you know, trying to calm down'. She'll be like [puts on a bored child's voice] 'OK'. . ... I mean it's a bit of everything. I think, I think me staying calm has made her calm as well. Um, obviously because if she's in a, in a panic and stressed out and I'm stressed out as well, you know, we'll be two headless chickens running 'round like, you know. Um, whereas when she's sort of really like, you know, frustrated and angry and sort of screaming and shouting and I just sort of sit there and say, 'you know, it's OK, it's OK to be angry, it's OK to be frustrated and, you know, if you want to talk to me, I'm here'. (Pamela—2.1)

Describing how she was learning to be calmer, Abbie remarked that she had previously being "scared" to go grocery shopping with her son, adding that this changed when she tried some strategies that her IY facilitator had suggested. Abbie believed her use of these strategies had helped her son feel more empowered during the shopping trips and helped her feel calmer. She excitedly described what a positive experience shopping had become for both of them, so much so that they were now going out to more places. Her fear seemed to have given way to calm confidence:

> I'm calmer because knowledge is power, isn't it? So you've got the knowledge that if you're calm, they're going to be calm. . ... If you're distracting them with a shopping list, they're going to have a pen and paper in their hand. So instead of just following mum, he's in control. . ... because kids generally feel like they're not in control of anything in their life. . ... And he's walking around 'Oh yeah we need this!' [referring to child reading from the list]. . ... So now we do restaurants, we do shopping, we do everything. I'm not scared. . ... So it gives the parent the confidence. (Abbie—1)

## Discussion

The analyses of parents' narratives in this article are not intended to allow conclusions to be drawn about the causal mechanisms involved in parenting programmes, but rather to provide a parent-centred perspective on the ways the programmes helped them. These analyses suggest that as parents' understanding of themselves and their children developed during parenting programmes they became more motivated to change. Sometimes parents seemed to experience "light bulb" moments of understanding arising from the programmes, but mostly parents perceived their understanding developed through gradual reflection on themselves and interactions with their child. While the behaviour of some children continued to be challenging, parents perceived that the programmes helped them feel calmer about the child's behaviour and more confident about their parenting. The narratives also provide insights on how parents made the changes occur in their everyday lives. When they felt calmer and more confident, parents perceived they were better able to introduce changes to their parenting, and crucially, to keep going if the changes they made did not initially seem to help. In this way, while their child's behaviour may have been the impetus for the parent being referred to the parenting programme, these positive transformations came from a deeper understanding of how different parents and children could positively interact with each other.

The current findings resonate with Aujoulat's emphasis on value of self-reflection in enabling a person to "let go" of both their own and others' presumptions and expectations [17] (p. 1236). Applying this to the parents in this study, the programmes seemed to support parents to 'let go' of anxiety they had about controlling their child's behaviour, helping parents to translate the insights they had gained about themselves and their children into responses that were more flexible. In turn, parents described gaining confidence and experiencing interactions with their child as more positive. For some parents letting go meant allowing their child to take on new responsibilities, which in turn helped to strengthen their relationship with their child and decrease the parent's own anxiety. For other parents, being able to 'let go' of feelings of responsibility around some of their children's challenging behaviours allowed parents to better distinguish behaviours that their children struggled to control, or between behaviours that were merely undesirable and those that were dangerous, enabling parents to respond in ways that were more attuned to their children. In line with Doubt and colleagues [12], as parents were less fatigued by the constant emotional labour and anxiety associated with trying to control their child's behaviour, they had space to further reflect and refine their understandings, creating a pathway to further positive change.

Differences in perceptions of their child's behaviour could be seen as 'small changes' [16], but these were important changes, especially for parents in Group 2.1 (positive experience of IY and perception of child behaviour but negative or no quantitative change in child behaviour). As confidence grew among these parents, so did their sense of calm. This supports Seabra-Santos and colleagues' [3] argument that IY helps to increase parents' feelings of efficacy which can then lead to changes in how a parent interacts with their child and improved child behaviour. However, many of the participants in the current study who attended other parenting programmes similarly described feeling calmer, suggesting that improved efficacy has a role in the positive changes in parent-child interaction seen with other types of parenting programmes and not just IY.

Mouton and Roskam [27] argue that there can be almost immediate positive and tangible changes in the way a parent interacts with their child after the parent perceives positive reinforcement. In line with this, for many parents in the current study, positive reinforcement came from seeing their child's behaviour improve or by feeling more confident and calmer in themselves about their child's behaviour. Encouragement from facilitators was also, at times,

the impetus that seemed to help parents to reframe their experiences as little accomplishments rather than hopelessness [16,17]. The impact of these types of positive reinforcement were illustrated in the examples parents from Groups 1 (positive experience of IY and positive change in child behaviour) and 2.1 gave of learning how to talk about feelings with their child or recognising the behaviours their child could not control. In contrast, despite some parents from Groups 2.2 (negative experience of IY and no improvement or worsening of child behaviour) and 3 (dropped out and did not attend) reporting positive changes in how they felt about their child's behaviour, none reported encouragement from the facilitators at the recent IY programme. Nevertheless, some believed they had found this encouragement in previous parenting programmes. Positive reinforcement, whether from seeing their child's behaviour improve, feeling calmer or from facilitators, appeared to help parents to engage with the parenting programme even if they had been hesitant to start with.

The reported absence of encouragement from the facilitators as noted by parents in Groups 2.2 and 3 highlights the importance of attending to the differences between how programmes are intended to be delivered and how they are experienced by parents. It is all the more important to continue to dissect the factors within a parenting programme that lead to positive and negative outcomes for the parent, and consequently the child, given the implications for the future wellbeing of both. When families of children who present with conduct problems are left unsupported, children tend to have a lower quality of life in adulthood compared to children without such diagnoses [28]. The wellbeing of parents who repeatedly experience poor services for themselves, and their children, is also negatively impacted and they are less likely to continue seeking help [4]. This study did not involve direct observations of programmes being delivered, and so we do not know how far the narratives of parents, including those in Groups 2.2 and 3, reflect what actually happened in sessions. However, the reported lack of facilitator encouragement from parents in Groups 2.2 and 3 suggests that at best, encouragement was not provided in ways that were meaningful for these parents. More broadly, our findings also demonstrate the importance of parents having agency within a parenting programme [18] where they feel enabled to explore, with guidance and encouragement from facilitators, what strategies and approaches fit within their own family dynamic.

Doubt and colleagues [12], (p. 18) argue that, when parents struggle with their children, parenting can become just "a continued effort towards survival". The narratives of Group 1 parents provide insights on how parenting can become less of a struggle following a parenting group, as their child's behaviour improved. Group 2.1 parents though provide an interesting comparison as their perception of themselves and their children had changed even if their child's behaviour had ostensibly not. Parents' perceptions of having benefitted from a parenting programme, without an accompanying change in their child's behaviour, at least as measured quantitatively, is worth further investigation. While the parents' narratives in this paper were positive, it is important to monitor the child's behaviour to ensure that they are also able to receive any appropriate support in the future. 'Calmer parent, calmer child' may have been enough to satisfy the parent in the short-term, but we do not yet know whether that leads to better long-term outcomes for the children themselves. Future quantitative evaluations of parenting programmes could look at involving parents, as well as health professionals and other stakeholders, in determining what outcomes are prioritised to assess effectiveness and the magnitude of change required for a programme to be deemed effective.

## Limitations

This study has some limitations. Given that the data were collected in the UK, it is open to question how far the findings are transferable beyond this context. While we sampled parents

from areas that are socio-economically and ethnically diverse, did not collect data on parents' socio-economic and ethnic characteristics. Future work exploring links between these characteristics and parents' narratives of change could inform future work to ensure programmes resonate with the diverse communities that they aim to serve. Most participants were women, and most of those mothers, but this reflects the people seeking support with parenting programmes. More male children were involved in the wider PPC programme, but this is in line with higher rates of diagnoses of conduct disorder among boys. Time only allowed for one interview so the findings offer a snapshot of experiences, and limits our knowledge as to causality or whether the positive transformations described were maintained over time.

## Conclusion

The findings suggest that when parents perceived that parenting programmes helped them to better understand themselves and their child, this led to more positive parent-child interactions, and a more empathetic relationship. While the challenging behaviour of some children did not lessen, at least according to quantitative measures, parents' narratives indicated that their feelings about the behaviours had changed, which seemed to help parents feel calmer. From the perspective of parents, programmes that enabled them to reflect on themselves and their responses to children helped them to incorporate what they had learnt into their parenting, and to improve their lives regardless of whether the child's behaviour improved. Facilitators of future programmes should look to see how their practice could better support reflection by parents and potentially help more families to benefit from programmes.

## Acknowledgments

The authors would like to thank Dr Rob Senior, Professor Stephen Scott, Professor Jonathan Hill, Dr Moira Doolan, Dr Matt Woolgar, and Siofra Peeren for their work on the Personalised Programmes for Children study.

## Author Contributions

**Conceptualization:** Kathy McKay, Eilis Kennedy, Bridget Young.

**Formal analysis:** Kathy McKay.

**Funding acquisition:** Eilis Kennedy, Bridget Young.

**Investigation:** Kathy McKay.

**Methodology:** Kathy McKay, Bridget Young.

**Supervision:** Eilis Kennedy, Bridget Young.

**Writing – original draft:** Kathy McKay.

**Writing – review & editing:** Kathy McKay, Eilis Kennedy, Bridget Young.

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
