## [Decision Letter · Decision Letter 0]

20 Oct 2020

PONE-D-20-20976

“Sometimes I think my frustration is the real issue”: Parents’ experiences of transformation after a parenting programme

PLOS ONE

Dear Dr. McKay,

Thank you for submitting your manuscript to PLOS ONE. After careful consideration, we feel that it has merit but does not fully meet PLOS ONE’s publication criteria as it currently stands. Therefore, we invite you to submit a revised version of the manuscript that addresses the points raised during the review process.

We look forward to receiving your revised manuscript.

Kind regards,

Andrea D. Warner-Czyz, Ph.D.

Academic Editor

PLOS ONE

Journal Requirements:

Additional Editor Comments (if provided):

Both reviewers find strengths in this manuscript, but it requires considerable reworking relative to justification of the rationale for this study (i.e. supporting literature, organized introduction), description of the methods, and conclusions based on the results. Both reviewers provide a plethora of feedback to benefit this paper.

Reviewers' comments:

Reviewer's Responses to Questions

**Comments to the Author**

1. Is the manuscript technically sound, and do the data support the conclusions?

Reviewer #1: No

Reviewer #2: No

2. Has the statistical analysis been performed appropriately and rigorously? 

Reviewer #1: No

Reviewer #2: Yes

3. Have the authors made all data underlying the findings in their manuscript fully available?

Reviewer #1: Yes

Reviewer #2: No

4. Is the manuscript presented in an intelligible fashion and written in standard English?

Reviewer #1: Yes

Reviewer #2: Yes

5. Review Comments to the Author

Reviewer #1: Summary of the Research

The manuscript used a qualitative approach to analyze themes in parent interviews regarding parent perceptions of participating in an Incredible Years (IY) parent training program. The authors claim that parents improved their understanding of their children and themselves after participating in parent training. Additionally, the authors claim that participants in the first two groups reported feeling calmer and more confident as parents.

Strengths Overview

The manuscript highlights the perceptions of parents who did or did not attend a parenting program. The introduction highlights the importance of understanding parents' motivation for change and positions the study as a mechanism for understanding parents' motivation for change during and after participation in a parent training program. Using a qualitative approach to inquiry allows for a rich and in-depth exploration of the phenomena in context. The manuscript contains several relevant quotations to support the arguments of the authors. The topic of the current study is relevant to the current literature and could help expand the current knowledge about the perceptions toward parent training programs of parents who dropped out, did not attend, or whose child behaviour scores did not improve after completing the IY parent training curriculum.

Weaknesses Overview

The manuscript has several limitations that should be addressed. The manuscript lacks a clear research objective as the description is vague. The authors should clarify the following sections to avoid confusion. Additionally, the authors make causal claims that are not appropriate given the qualitative nature of the study.

Examples and Evidence

Given that the manuscript does not include line numbers, the feedback provided in this section is organized by header section. This reviewer used page numbers to describe the location of examples and evidence.

Abstract

1. The problem under investigation is not described in the abstract.

2. Clarification as to the type of parent programme (i.e., behaviour management training) would be beneficial in the abstract.

3. The authors should clarify whether the causal attribution made in the second to last sentence is referring to the parent's perception of the problem or study findings. The wording in that sentence suggests that this causal relationship was demonstrated in the study.

Introduction

4. The introduction nicely explains that positive parent-child interactions are beneficial for children and their parents. However, the link between positive parent-child interactions, positive parenting (mentioned on p. 3, first sentence of second paragraph), and parenting programs is not clear in the text. Consider explicitly stating the link between these variables in order to bring context to your study.

5. The second sentence of the first paragraph on page 3 connects two thoughts with the word "whereas." However, the wording of the two thoughts does not make them seem contradictory. Clarification is needed to understand the idea that the authors are attempting to convey in this sentence.

6. A brief description of parent training programmes in general would better frame the context of the study.

7. In addition to your review, consider providing a critique or synthesis of the applicable literature to identify key issues in the relevant literature and specify knowledge gaps in current literature. In other words, make it very clear that your study will fill in a gap in the literature after describing the current gap in knowledge.

8. There are no clearly stated research questions and the study objective is vague. After reading the introduction, I was still unsure as to what exactly the study was directly aiming to accomplish. Without specific research questions or objectives, one cannot evaluate if the approach was appropriate for addressing the research objectives.

9. Qualitative research allows for insight into the perception of participants. Consider providing a brief rationale for using a qualitative approach to address the research objectives.

Methods

10. Provide a rationale for fit of design used to investigate the study purpose.

11. In the first paragraph of Page 7, the manuscript describes the different parent groups and states that the groups were based on "children who showed improved behaviour" and "children who showed worsening behaviour." It is unclear how the child's behaviour was measured. The wording suggests that the change in behaviour was demonstrated by an observation ("showed") but it should be made clear. Although this change in behaviour is not the main variable of interest in the manuscript, a brief description would be helpful to understand the context of the study.

12. Given that generalizability to large populations is limited in a qualitative study, it would be helpful for the reader to understand a little more about the participants' demographics and cultural information.

13. When describing the sample, it would be helpful to be specific about the time at which this information was gathered. For example, in the first paragraph of Page 7, the manuscript describes the relational status of the parents. However, it refers to fluctuating relationship statuses over the course of the project, so it is not clear at which point the authors are describing the sample. At the time of the start of the project, during the project, or at the time of the interviews?

14. It would be beneficial for the reader to understand the specific purposive selection process method used and inclusion/exclusion criteria.

15. Briefly describe the coders or analysts and their training.

Results

16. The authors should clarify whether the causal attributions made in the results section are referring to the parent's perception of the problem, empirical research findings, or author's interpretation of study findings. The wording of some sentences suggests that causal relationships were demonstrated in the study. Consider adding a citation to causal statements, including evidence for the statement, or rewording the sentences to make them more accurate. A list of some causal statements follows:

○ Page 13, first paragraph: "By helping parents to ground their parenting in empathy, IY and other parenting programmes changed the way these parents interacted with their children."

○ Page 17, second paragraph: "Kindness could change every-day interactions with children too."

○ Page 20, first paragraph: "Parenting programmes helped to remind other participants that their children could not necessarily understand things or act in the same way as adults."

Discussion

17. The authors should describe how the current study findings are similar and different from prior research findings.

18. Consider describing the types of contributions made by the current study findings in relation to previous research (e.g., challenging, elaborating on, or supporting prior research).

19. Consider describing how the study findings can be utilized by practitioners and or researchers.

Limitations and Conclusion

20. The authors claim to demonstrate that there is a causal relationship between a parent's increased understanding of themselves and their child and an increase in parent-child interactions (see p. 28, first sentence of second paragraph in the Discussion section). The authors state: "The findings indicate that when parenting programmes helped parents to better understand themselves and their child, this led to more positive parent-child interactions, and a more empathetic relationship." However, the data and methods do not fully support this conclusion. Specifically, a qualitative study cannot prove the causal influence of one variable on another. Additionally, the variables in this sentence are not defined. For example, "a parent's increased understanding of themselves" indicates that there is an increase from a baseline measurement of parental understanding of themselves. The question of how the authors know that understanding and empathy increased from before the program started is left unanswered. Understanding and empathy are difficult constructs to measure. Parent-report of these constructs should be clearly identifiable as such.

21. The last sentence of the manuscript is confusing which limits the readability of your conclusion.

References

22. The authors should review the References section of the PLOS ONE Submission Guidelines and the NIH website with International Committee of Medical Journal Editors (ICMJE) Samples of Formatted References (https://www.nlm.nih.gov/bsd/uniform_requirements.html) for information regarding the references. Specifically, the authors should:

○ Place a colon after the volume number

○ Format the page numbers according to the guidelines (e.g., remove "pp." on page 30)

○ Remove the extra year from the fifth reference on page 30

Grammatical and Stylistic Review

1. The first letter in section heading words should be capitalized (i.e., "Limitations and conclusion" header on page 28).

2. Some of the sentences are wordy, making it difficult to follow. I advise the authors work to improve the flow and readability of the text (e.g., run-on sentences should be reworded or separated into two sentences, replace wordy sentences with concise statements, etc.). For example, the last sentence of the second paragraph on page 4 can be condensed by removing the first six words of the sentence. There is no need to mention the reference twice in the same sentence. The additional citation takes away from the readability of the text.

3. Consider replacing "our study" with "the current study" throughout the document.

4. Consider replacing “we” with “the researchers” throughout the document.

Reviewer #2: This study explores the perspectives of participants / nonparticipants in an Incredible Years (IY) program. The data were collected in conjunction with a larger multi-phase, mixed methods study which is ongoing. As part of this qualitative study, 42 participants were interviewed about their experiences and the data subjected to analysis through a phenomenological lens. This is an impressive sample size in a qualitative study. Three groups of participants were distinguished: (1) participants in IY programs who reported improved behaviour in their child, as indicated on a quantitative measure (Parent Account of Child Symptoms; PACS); (2) participants in IY programs whose children showed no change or worsening behaviour as measured by the PACS; (3) nonparticipants or participants who had dropped out of the IY program.

As the authors note, the perspectives of parent/ caregiver participants are an important source of information about parent groups designed to address and, hopefully, improve behavioural concerns in their children. In the paper, a number of compelling and informative quotes from participants in the study are included as part of the data reported, and attest to how powerful information gleaned from qualitative research can be. The size of the sample in the study and the detail in some aspects of the methodology reported are strengths of the paper. For example, details about the interviewers and the interviews including the use of recordings, transcription and field notes are noteworthy and consistent with SRQR and COREQ standards utilized by PLOS in evaluating qualitative research. I would recommend that the specific questions used to guide the interviews also be included in the paper for further information. In addition, it might be helpful to include the background and training of the interviewers as well, given the clinical nature of the programs which are being explored. In addition, a general description of the settings in which the programs are delivered (e.g., community, school, hospital) might be helpful.

However, I have the following general comments about information presented in the paper.

In the Introduction, the rationale for conducting the study needs to be stated more explicitly. It is not clear from the Introduction why the study was conducted and what the qualitative research methodology contributes to that research question. Interesting research in the area is reviewed but it needs to be integrated, and how this research relates specifically to the study needs to be more explicitly stated. There are references to the findings and perspectives of several researchers and it would be helpful to describe the specific nature of their contributions and how these contributions informed the rationale and interpretation of the findings of the current study. In addition, in the Introduction, there should be a description of IY programs, including goals, content and process of program delivery.

In the design, as presented, all participants were recruited as part of participation in the IY programs, although participants in Group 3 either did not complete, or attend, the programs. The IY program is an important reference in the design and needs to be included in the interpretation of results - understanding that parents may often access multiple resources (as noted for Group 3): participants were recruited from them, and 34 participants did attend IY programs. There was also little discussion about why many caregivers in Group 2 reported improvements in their children’s behaviour, which was not seen on the quantitative measure. There are a number of potential explanations for this, which should be explored. For example, this difference could have implications from how change is measured or how outcomes are defined in future research, and speak to information which qualitative research can provide in understanding change.

In the results section, 3 themes are presented: Parents better understanding themselves; Parents better understanding their children and Calmer parent, calmer child. The experience of parents is described as ‘transformative’ (page 10), and it would be helpful if this construct were defined before themes are presented. Within each of these themes, presentation of the subthemes, and how they relate to the main theme needs to be made more explicit so that the theme is clear to the reader. In addition, the relationship of each quote to the theme needs more clarity.

In the discussion, the main findings of the study need to be stated clearly, and their implications discussed. Some interesting work is presented, but is not described in sufficient detail, and its relationship to the findings in the current study is not made clear. In addition, the relationship of the findings of the study to the goals of the IY parenting programs is not discussed. While I note references to several parenting strategies in the quotes provided by participants, there is no discussion of strategies / program content or their potential implication for parents’ experiences as group participants.

Minor comments for consideration are:

The format of the abstract needs to be revised

In the Method section, information about the Personalized Programmes for Children (PPC) Project includes a description about design, but is limited in any information about content: is this study the qualitative phase of the early phase of the larger project?

More information about scores on the PACS which formed criteria for inclusion in Group 1 vs Group 2 should be stated

If reasons to decline participation were given, it would be helpful to include these reasons

Note that 42 participants were reported, although the subcategories add up to 41

I appreciate the inclusion of participants who were not biological parents. However, it would be helpful to include some (non-identifying) information about the role these individuals played in children’s lives e.g., primary caregiver in the home; primarily caregiver on a daily basis etc.

Some information included in the Ethics section (e.g., contact with Research Assistants, followed by contact with Qualitative Interviewers) should also be included in the Method section

6. PLOS authors have the option to publish the peer review history of their article (what does this mean?). If published, this will include your full peer review and any attached files.

Reviewer #1: No

Reviewer #2: No

---

## [Author Response · Author response to Decision Letter 0]

4 Dec 2020

'Response to Reviewers' document has been attached with all the changes listed as required by the reviewers.

---

## [Decision Letter · Decision Letter 1]

15 Jan 2021

PONE-D-20-20976R1

“Sometimes I think my frustration is the real issue”: Parents’ experiences of transformation after a parenting programme

PLOS ONE

Dear Dr. McKay,

Thank you for submitting your manuscript to PLOS ONE. After careful consideration, we feel that it has merit but does not fully meet PLOS ONE’s publication criteria as it currently stands. Therefore, we invite you to submit a revised version of the manuscript that addresses the points raised during the review process.

We look forward to receiving your revised manuscript.

Kind regards,

Andrea D. Warner-Czyz, Ph.D.

Academic Editor

PLOS ONE

Additional Editor Comments (if provided):

We appreciate the efforts of the author to address feedback from reviewers in this revision of a project covering an important topic - the role of a targeted program to support parents of children with aggression or related conduct problems. However, some issues remain unresolved (i.e., need for greater organization and clarity in methods and results; lack of synthesis in the discussion section; need for careful proofreading for spelling, grammatical, and punctuation errors and readability). The Additionally, Reviewer 2 raises a crucial point in the need for the author to differentiate this paper from content and analyses in previously published papers using the same data set.

Reviewers' comments:

Reviewer's Responses to Questions

**Comments to the Author**

1. If the authors have adequately addressed your comments raised in a previous round of review and you feel that this manuscript is now acceptable for publication, you may indicate that here to bypass the “Comments to the Author” section, enter your conflict of interest statement in the “Confidential to Editor” section, and submit your "Accept" recommendation.

Reviewer #1: (No Response)

Reviewer #2: (No Response)

2. Is the manuscript technically sound, and do the data support the conclusions?

Reviewer #1: Yes

Reviewer #2: No

3. Has the statistical analysis been performed appropriately and rigorously? 

Reviewer #1: Yes

Reviewer #2: Yes

4. Have the authors made all data underlying the findings in their manuscript fully available?

Reviewer #1: Yes

Reviewer #2: Yes

5. Is the manuscript presented in an intelligible fashion and written in standard English?

Reviewer #1: Yes

Reviewer #2: No

6. Review Comments to the Author

Reviewer #1: Strengths Overview

• The topic expands current knowledge and adds to the body of literature on parent perceptions of their parenting and children following parent training programmes. The authors did a nice job of implementing the suggestions from the prior review in their updated manuscript. The readability, organization, and cohesiveness of the paper greatly improved, making a much clearer and stronger argument for their findings. The description of the study design was thorough and helped me better understand the methods. The description of themes and highlighted participant quotes are very appropriate and greatly add to the paper. The added details to the discussion section are also helpful for interpreting the results in context.

Weaknesses Overview

• Minor stylistic and readability issues are discussed here.

Minor Issues

• The new subheadings in the results section are very helpful. However, the style of the second- and third-level headings are the same, which is confusing. Consequently, it seems like there is missing text under lines 325 and 509.

• Ensure consistency in citation and quotation style (e.g., punctuation differences in lines 117 and 261).

• Double citations in the same sentence are redundant and limit readability of the text (e.g., line 123)

Reviewer #2: This study explores the perspectives of participants / nonparticipants in parenting programs and utilizes a qualitative research methodology. This paper is a revision of an earlier submission, and responses to the comments of 2 reviewers have also been provided. The strengths of qualitative approaches to this topic noted in previous review continue to be relevant. In general, while responses to reviewer comments to address weaknesses were made, more precision and detail is needed in key areas. In addition, comments about clarity and about grammar continue to be relevant.

Major Weaknesses:

1. Context of the Study:

A previously published qualitative study of this phase of the research is referenced in the Method section (McKay et al., 2020): based on the information provided in this submission as well as the McKay et al (2020) paper, these two reports appear to be based upon the same interviews. If so, differentiation and/or integration of information in this paper with that of the recently-published is needed.

2. The research question, as stated, remains unclear:

On page 6, the questions are stated as: to examine “how and why parents felt motivated to change” and “how parents made changes occur”

On page 7, this study “broadly sought to improve understanding of how programmes led to good outcomes for some parents, and poor outcomes for others from the perspective of parents”

It would be helpful to clarify what is meant by “change” and what is the object of study. Is change defined by the child’s behaviour from which the groups were purposively selected (e.g. on the PACS? Is change defined by parents as a result of participation (or not) in the parenting groups (if so, how was this defined?)? Or is change defined by parents’ perceptions of their own motivation? Or the changes they made after participating in the group (or not)?

I think this lack of clarity in the research question contributes to lack of clarity in presenting and interpreting the results. It also contributes to confusions which arise in understanding the causal statements made in the discussion (previous review; see also below).

3. Clarification of Results (made in the previous review) not addressed

Additional Areas to be Addressed:

4. Introduction

It would be helpful to provide some background as to the content of parenting programs, including IY, in the Introduction as the goals and issues to be addressed may differ between programs. That information provides a context from which parents’ experiences and perceptions are derived and has the potential to inform interpretation. For example, attributes of parent knowledge and skill are described in the first sentence of the Introduction, although strategies is a components referred to later on in the Intro.

Positive parenting is an approach which is not sufficiently described in the Introduction: it is not clear in paragraphs 2 and 3 whether the outcomes described as for positive parenting approaches or for components of parenting programs.

5. Method

Recruitment process still unclear for some participants (namely Group 3): if they were referred, or considered being referred, but did not attend coffee sessions, how were they recruited?

1 participant who gave permission to be contacted but was not interviewed is not accounted for in description of participants

Information about the PACS and about criteria from the PACS used to select participants is not specified. This was previously identified as a concern, and is not sufficiently addressed in the revision.

In addition, were these children identified with ‘severe behaviour problems’ — in that on page 11, the authors indicate that the interviewers did not have personal experienced of parenting children with severe behaviour problems, which allows that interpretation to characterize the patient sample.

Providing the interview questions and follow up probe question has been strongly recommended and would help further clarify how the research question was operationalized. In addition, were participants asked questions about the phenomena of interest (i.,e from page 6: how and why parents felt motivated to change, how parents made changes occur) or was this inferred by parents’ responses to other questions

The relationship of participants to the children, specified on page 11, is helpful. It should also be placed earlier (e.g. page 9 line 206) when recruitment of participants is discussed. I suggest that use of the term ‘parent’ may be misleading, given the sample, and would suggest a term (e.g., parent/ caregiver) to indicate the broader roles that the participants appear to have in the lives of children.

6. Ethics:

Participants’ comments about the cathartic nature of participation should not be in this section. It is, however, interesting and if it emerged as a theme, might be considered to be included in the results section, possibly mentioned there.

7. Results:

I would suggest some reorganization to reduce confusion and enhance clarity: Although it is not stated explicitly, my inference is that the analysis was conducted on all interviews and themes emerged irrespective of subgroup. If so, I would suggest presenting the themes first, and include a separate section looking at differences between samples.

A description of the 3 superordinate categories is missing, which seems to be the way in which the following comment was addressed from the previous review — this section still needs clarification. In the results section, 3 themes are presented: Parents better understanding themselves; Parents better understanding their children and Calmer parent, calmer child. Within each of these themes, presentation of the subthemes, and how they relate to the main theme needs to be made more explicit so that the theme is clear to the reader. In addition, the relationship of each quote to the theme needs more clarity.

There continues to be a need for clarity in presenting causality both in the Results and in the Discussion.

8. Discussion:

In the discussion, the results should be summarized and then interpreted. It is difficult to ascertain findings vs. interpretation.

There is no discussion of the differences between the different samples reported in the results section, findings which should be addressed.

There is no discussion of the finding that within Group 2, a major subgroup reported positive changes in children’s behaviour which were not apparent on the qualitative measure. While the authors’ response is that this finding is outside the scope of the study, they do report it as a finding, and it needs to be discussed.

9. The article continues to need proofreading for grammar and for readability.

7. PLOS authors have the option to publish the peer review history of their article (what does this mean?). If published, this will include your full peer review and any attached files.

Reviewer #1: No

Reviewer #2: No

---

## [Decision Letter · Decision Letter 2]

20 Apr 2021

PONE-D-20-20976R2

“Sometimes I think my frustration is the real issue”:

A qualitative study of parents’ experiences of transformation after a parenting programme

PLOS ONE

Dear Dr. McKay,

Thank you for submitting your manuscript to PLOS ONE. After careful consideration, we feel that it has merit but does not fully meet PLOS ONE’s publication criteria as it currently stands. Therefore, we invite you to submit a revised version of the manuscript that addresses the points raised during the review process.

We look forward to receiving your revised manuscript.

Kind regards,

Andrea D. Warner-Czyz, Ph.D.

Academic Editor

PLOS ONE

Journal Requirements:

Additional Editor Comments (if provided):

The reviewers and I agree the authors improved this paper from its original form based on feedback from previous reviewers. However, we feel the manuscript needs to address additional issues - both conceptual and line-level - before becoming suitable for publication.

Reviewers' comments:

Reviewer's Responses to Questions

**Comments to the Author**

1. If the authors have adequately addressed your comments raised in a previous round of review and you feel that this manuscript is now acceptable for publication, you may indicate that here to bypass the “Comments to the Author” section, enter your conflict of interest statement in the “Confidential to Editor” section, and submit your "Accept" recommendation.

Reviewer #1: (No Response)

Reviewer #3: (No Response)

2. Is the manuscript technically sound, and do the data support the conclusions?

Reviewer #1: Yes

Reviewer #3: Yes

3. Has the statistical analysis been performed appropriately and rigorously? 

Reviewer #1: Yes

Reviewer #3: N/A

4. Have the authors made all data underlying the findings in their manuscript fully available?

Reviewer #1: Yes

Reviewer #3: No

5. Is the manuscript presented in an intelligible fashion and written in standard English?

Reviewer #1: Yes

Reviewer #3: Yes

6. Review Comments to the Author

Reviewer #1: Strengths Overview

The topic expands current knowledge and adds to the body of literature on parent perceptions of their parenting and children following parent training programmes. The authors did a nice job of implementing the suggestions from the prior reviews in their updated manuscript. Overall, the structure and wording has greatly improved.

Suggestions

While the authors did a nice job of strengthening the manuscript with suggestions from the previous review (e.g., expanded upon what the IY program is about), there are some minor concerns that should still be addressed.

1. Line 120: Reword “learn what and how to change about their parenting” part of the sentence.

2. Line 136: Reword “how people adjust to the changes to their lives” to “life changes” in order to help the readability of the sentence.

3. Lines 154-155: Please update the description of the IY program as the IY program is not founded upon “the premise that negative reinforcement by parents develops and maintains negative behaviours in children.” Actually, in addition to positive reinforcement, the IY program encourages parents to negatively reinforce appropriate behaviors of their children (e.g., asking politely to stop an activity, avoiding strangers, etc.). Negative reinforcement (i.e., allowing a child to avoid or escape an activity) is not the same as punishment.

4. Lines 267-270: consider describing the interviewers/researchers earlier in the text before referencing them. In other words, some of the information in this sentence should be presented before the interviewers are mentioned.

5. While the authors did a very nice job of making the writing clearer and more concise, there are still some sentences with multiple embedded clauses that interrupt the flow.

o Consider giving examples in parenthesis or breaking the sentence into two.

o Also, some of the same sentences that have multiple embedded clauses, also have missing punctuation, which makes it difficult to follow. For example, Lines 498-500 “At the time she was interviewed Olivia…”

6. The punctuation/wording of some sentences is not grammatically correct. For example:

o Line 338-339: “In the sections below, while we…”

o Line 381-383: “Bea, a Group 1 participant…”

o Line 567-568: “Several children in the study…” does not need a comma.

o Lines 834-836: “The findings indicate…”

Reviewer #3: In relation to one of the questions above: The data (from qualitative interviews) is not made publicly available to protect confidentiality of data related to family lives and environments. The data may be accessed however on reasonable request addressed to the authors, which is fine

Other comments addressed to the authors:

This is very nice piece of work, which leads the reader to wanting to know more about the ambitious mixed-methods research project described.

Regarding the specific part reported in this manuscript, I have a few questions in relation to the methods section, and two suggestions for the discussion :

My first question refers to the choice of having a group of respondents (group 3) who had dropped out or not attended an YI programme at all? As the objective of the study was to explore the changes reported by parents in relation to attending a programme, the choice of recruiting parents who had not attended a programme does not seem completely justified. I wonder if it would be enough to focus on the narratives from the parents of groups 1 and

2.

My second question is about the 200 pounds reward/incentive to participate in the study. It is not entirely clear to me if this was to participate in the qualitative interviews only or for the whole study. From my own experience of conducting research it seems a lot of money, and I would like to invite the authors to give a bit more details: how and when this was presented to the parents, why they think this was justified or necessary, how they think this might have influenced participation.

Unless I missed the information which might be presented as an appendix elsewhere (?), I did not see many details about the sample. In particular, some socio-economic or cultural characteristics would be useful, as well as some information regarding the health status of the participating parents (if relevant... it is mentioned in the results that one participant had had depression, which surely influences how child behaviours mightbe perceived), and whether they were professionally active when participating in the programme and later in the interviews. I understand that the sample was purposively diversified. It would be interesting to know if and how the authors looked at possible influences of some personal characteristics on the benefits associated by the parents with their participation in the programme.

Last: one interview lasted only 16 minutes. It would be interesting to know a bit more about why this interview was so short and why the authors still considered it worthwhile to include it in their dataset.

Regarding my two suggestions for the discussion:

- I struggled a bit from the beginning about the presentation of parenting programmes aimed at changing child behaviours, and was wondering why the primary outcomes of such programmes should be child behaviours rather than parental behaviours, expectations, self-representations, etc., as well as parent-child interactions. It is clear from the results and later in the discussion that parental changes are among the main outcomes reported by the parents, and that changes in their perceptions of their child behaviours might be as important as the objective behaviours, or even more important. I think the assumption from the introduction that parenting programmes would change child behaviours could be further critically addressed in the discussion

- Also, I would welcome some critique of positive parenting programmes as they are also believed to increase the risk of parental burn-out, in shaping models of an ideal parent, and consequently discrepancies between self-representations and ideal representations of a "good parent". While the authors acknowledge that the aim of their article was not to look at causal mechanisms that would support positive changes, it would yet be interesting to go a bit more into details of key-ingredients of such programmes that are believed to play a positive role. The authors could build on their suggestions that self-reflection and learning to let go are essential, but what are the components of a programme that seem key to achieve this?

7. PLOS authors have the option to publish the peer review history of their article (what does this mean?). If published, this will include your full peer review and any attached files.

Reviewer #1: No

Reviewer #3: **Yes: **Isabelle Aujoulat

---

## [Author Response · Author response to Decision Letter 2]

3 Jun 2021

'Response to Reviewers' document has been attached with all the changes listed.

---

## [Decision Letter · Decision Letter 3]

30 Jun 2021

PONE-D-20-20976R3

“Sometimes I think my frustration is the real issue”:

A qualitative study of parents’ experiences of transformation after a parenting programme

PLOS ONE

Dear Dr. McKay,

Thank you for submitting your manuscript to PLOS ONE. After careful consideration, we feel that it has merit but does not fully meet PLOS ONE’s publication criteria as it currently stands. Therefore, we invite you to submit a revised version of the manuscript that addresses the points raised during the review process.

We look forward to receiving your revised manuscript.

Kind regards,

Andrea D. Warner-Czyz, Ph.D.

Academic Editor

PLOS ONE

Journal Requirements:

Additional Editor Comments (if provided):

Reviewer 2 makes a very insightful comment re: conduct problems, quality of life, and the potential role of parent-child interactions. Please address this, as well as their comments about compensation.

Reviewers' comments:

Reviewer's Responses to Questions

**Comments to the Author**

1. If the authors have adequately addressed your comments raised in a previous round of review and you feel that this manuscript is now acceptable for publication, you may indicate that here to bypass the “Comments to the Author” section, enter your conflict of interest statement in the “Confidential to Editor” section, and submit your "Accept" recommendation.

Reviewer #1: All comments have been addressed

Reviewer #3: (No Response)

2. Is the manuscript technically sound, and do the data support the conclusions?

Reviewer #1: Yes

Reviewer #3: Yes

3. Has the statistical analysis been performed appropriately and rigorously? 

Reviewer #1: Yes

Reviewer #3: N/A

4. Have the authors made all data underlying the findings in their manuscript fully available?

Reviewer #1: No

Reviewer #3: Yes

5. Is the manuscript presented in an intelligible fashion and written in standard English?

Reviewer #1: Yes

Reviewer #3: Yes

6. Review Comments to the Author

Reviewer #1: The topic expands current knowledge and adds to the body of literature on parent perceptions of their parenting and children following parent training programmes. The authors did a nice job of implementing the suggestions from the prior reviews in their updated manuscript. The conceptualization, readability, organization, and cohesiveness of the paper greatly improved, making a much clearer and stronger argument for their findings. I especially enjoyed the quotes as they made the themes come to life. I would suggest merging the two sentences on lines 863-864 (“The data were collected…” because the way it is currently written makes it seem that data collected in the U.K. is a limitation. One example of how to change this wording could be, “Given that the data were collected in the U.K., it is open to…” Overall, the authors have greatly improved the manuscript to make it ready for publication.

Reviewer #3: All comments have been well addressed in the letter to the reviewers, and I thank the authors for engaging in the discussion around my comments. I still have two minor comments/suggestions in relation to two of these comments:

- regarding the compensation of 200 pounds, I would like to recommend that the full explanation given in the letter to the reviewers be added to the manuscript, starting from "Many parents we approached were from underserved communities" to "the money was a total surprise to them".

- regarding the characteristics of the sample : I understand the authors do not have socio-demographic details to share. The added sentence "Care was also taken to ensure a socio-economically and ethnically diverse sample, however data on socio-economic status or ethnicity were not collected in the qualitative study" does not make sense to me. If care was taken to ensure diversity, the information should exist. If it exists for the quantitative study (n=139 ?) maybe something could be said about this initial sample, and the authors could indicate wether they assume their qualitative sample shares more or less the characteristics of the quantitative sample. Or they do not mention it at all in the methods section and acknowledge it as a limit in the discussion, as it might have been interesting to look at the influence (even hypothetically) of some characteristics on the findings.

Last, it's a personal view, and I would not like to request from the authors that they change their own, but I find it a pity to end with the sentence "children whose conduct problems are left untreated (...)", when the results show that specific behaviours might be the product of dysfunctional parent-child interactions, and children behaviours might be perceived as problematic but not necessarily be problematic. The strength of a successful parenting program is according to me precisely to help parents change their views of themselves and their children, and consequently their interactions and communication with their children, which may in turn have a positive effect on children's behaviours.

7. PLOS authors have the option to publish the peer review history of their article (what does this mean?). If published, this will include your full peer review and any attached files.

Reviewer #1: No

Reviewer #3: **Yes: **Isabelle Aujoulat

---

## [Decision Letter · Decision Letter 4]

30 Sep 2021

“Sometimes I think my frustration is the real issue”:

A qualitative study of parents’ experiences of transformation after a parenting programme

PONE-D-20-20976R4

Dear Dr. McKay,

We’re pleased to inform you that your manuscript has been judged scientifically suitable for publication and will be formally accepted for publication once it meets all outstanding technical requirements.

Kind regards,

Andrea D. Warner-Czyz, Ph.D.

Academic Editor

PLOS ONE

Additional Editor Comments (optional):

Reviewers' comments:

Reviewer's Responses to Questions

**Comments to the Author**

1. If the authors have adequately addressed your comments raised in a previous round of review and you feel that this manuscript is now acceptable for publication, you may indicate that here to bypass the “Comments to the Author” section, enter your conflict of interest statement in the “Confidential to Editor” section, and submit your "Accept" recommendation.

Reviewer #1: All comments have been addressed

Reviewer #3: All comments have been addressed

2. Is the manuscript technically sound, and do the data support the conclusions?

Reviewer #1: Yes

Reviewer #3: Yes

3. Has the statistical analysis been performed appropriately and rigorously? 

Reviewer #1: Yes

Reviewer #3: N/A

4. Have the authors made all data underlying the findings in their manuscript fully available?

Reviewer #1: Yes

Reviewer #3: No

5. Is the manuscript presented in an intelligible fashion and written in standard English?

Reviewer #1: Yes

Reviewer #3: Yes

6. Review Comments to the Author

Reviewer #1: See below for a few grammatical/stylistic suggestions:

318: comma after services or replace "as well as" with "and"

318-320: consistency in verb tense (currently switches from past to present within the same sentence)

318-320: “… many of the parents may not feel they were not listened to or valued sufficiently…”

- I am confused by the meaning of this sentence. Should the “not” before feel in line 318 be removed?

Line 867-869: repetition of “future work” within the same sentence could be reconsidered

Reviewer #3: I think there is a likely error in the following sentence added to the revised version :

(line 319) We anticipated that many of the parents may not feel they were not listened to or valued sufficiently, and we wanted to create a safe space for them

I guess it should read "may feel" instead of "may not feel" ?

My comments were otherwise well taken into accunt. Thank you.

7. PLOS authors have the option to publish the peer review history of their article (what does this mean?). If published, this will include your full peer review and any attached files.

Reviewer #1: No

Reviewer #3: No

---

## [Editor Report · Acceptance letter]

4 Oct 2021

PONE-D-20-20976R4 

“Sometimes I think my frustration is the real issue”:
A qualitative study of parents’ experiences of transformation after a parenting programme 

Dear Dr. McKay:

I'm pleased to inform you that your manuscript has been deemed suitable for publication in PLOS ONE. Congratulations! Your manuscript is now with our production department. 

Kind regards, 

on behalf of

Dr. Andrea D. Warner-Czyz 

Academic Editor

PLOS ONE